# Sir2 and Reb1 antagonistically regulate nucleosome occupancy in subtelomeric X-elements and repress TERRAs by distinct mechanisms

**Stefanie L. Bauer**[1], **Thomas N. T. Grochalski**[1], **Agata Smialowska**[2], **Stefan U. Åström**[1]*

**1** Department of Molecular Biosciences, the Wenner-Gren Institute, Stockholm University, Stockholm, Sweden, **2** National Bioinformatics Infrastructure Sweden, Science for Life Laboratory, Stockholm University, Stockholm, Sweden

* stefan.astrom@su.se

**Data Availability Statement:** All MNase sequencing data have been deposited on Gene Expression Omnibus (GEO) under accession number GSE195972. MNase-qPCR data as well as

## Abstract

Telomere chromatin structure is pivotal for maintaining genome stability by regulating the binding of telomere-associated proteins and inhibiting the DNA damage response. In *Saccharomyces cerevisiae*, silent information regulator (Sir) proteins bind to terminal repeats and to subtelomeric X-elements, resulting in transcriptional silencing. Herein, we show that *sir2* mutant strains display a specific loss of a nucleosome residing in the X-elements and that this deficiency is remarkably consistent between different telomeres. The X-elements contain several binding sites for the transcription factor Reb1 and we found that Sir2 and Reb1 compete for stabilizing/destabilizing this nucleosome, i.e. inactivation of Reb1 in a *sir2* background reinstated the lost nucleosome. The telomeric-repeat-containing RNAs (TERRAs) originate from subtelomeric regions and extend into the terminal repeats. Both Sir2 and Reb1 repress TERRAs and in a *sir2 reb1* double mutant, TERRA levels increased synergistically, showing that Sir2 and Reb1 act in different pathways for repressing TERRAs. We present evidence that Reb1 restricts TERRAs by terminating transcription. Mapping the 5′-ends of TERRAs from several telomeres revealed that the Sir2-stabilized nucleosome is the first nucleosome downstream from the transcriptional start site for TERRAs. Finally, moving an X-element to a euchromatic locus changed nucleosome occupancy and positioning, demonstrating that X-element nucleosome structure is dependent on the local telomere environment.

## Author summary

Telomeres are specialized structures at the end of linear chromosomes that protect the genetic material from degradation and mistaken recognition as sites of damage. Telomere dysfunction has been linked to several diseases and senescence. The telomeres contain repetitive DNA sequences bound by specialized proteins. Here, we describe two such proteins, Sir2 and Reb1, which regulate the formation of nucleosomes at a repetitive sequence

MNase sequencing data used to generate the presented plots can be found at DOI 10.6084/m9.figshare.20527767.

**Funding:** SUÅ was supported by Cancerfonden www.cancerfonden.se (grant 20 1241 PjF 01 H) and Vetenskapsrådet, www.vr.se (grant 2015-05212). The funders had no role in study design, data collection and analysis, decision to publish or preparation of the mansucript.

**Competing interests:** The authors have declared that no competing interests exist.

known as the X-element. Sir2 has very important roles in regulating the accessibility of telomeres to the cellular machinery that reads and transcribes the genetic material. Reb1 had not been previously implicated in telomere biology, but is rather known as a general regulator of transcription. We explored the effects of removing either or both of these factors on telomeric features and their relationship in regulating the structure and accessibility of the telomeres in budding yeast. We show that Sir2 and Reb1 have opposing roles in stabilizing and de-stabilizing a nucleosome at the telomeres, but that both inhibit the accumulation of a non-coding RNA molecule transcribed from the telomeres.

## Introduction

Silent information regulator 2 (Sir2) is a conserved NAD$^+$-dependent histone deacetylase with homologues in Archaea, Eubacteria and most if not all eukaryotes [1–3]. In *Saccharomyces cerevisiae*, Sir2 targets lysine residues on both histone H3 and H4 [2,4,5], where deacetylation of lysine 16 (K16) on histone H4 is of particular importance [6]. *SIR2* was first found in a screen for mutations that derepressed the cryptic mating-type loci (*HMLα* and *HMR***a**) in *S. cerevisiae* [3]. Sir2 acts together with other Sir-proteins (Sir1, Sir3 and Sir4), establishing a repressive chromatin structure at the cryptic mating-type loci resulting in transcriptional silencing, despite presence of normal promoters. Subsequent studies revealed that Sir-proteins also silenced transcription at telomeres [7] and that Sir2 silenced transcription at the rDNA loci [8].

The telomeres protect chromosomes from exonucleolytic degradation, inhibit chromosome fusions and prevent a permanent DNA damage response that otherwise would lead to replicative senescence [9]. In *S. cerevisiae*, three different repetitive sequences characterize telomeres: terminal telomeric repeats, Y′-elements and X-elements [10], reviewed in [11]. The terminal repeats consist of ~300 (TG$_{1-3}$) sequences that are synthesized by telomerase. The terminal repeats contain multiple binding sites for the transcription factor Rap1, which recruits the Sir-proteins via protein-protein interactions [12]. Whereas the role of the terminal repeats in protecting chromosome ends is well characterized, the roles of the subtelomeric Y′- and X-elements are not well understood. About half of the *S. cerevisiae* telomeres contain Y′-elements, encoding a putative helicase. The X-elements consist of repetitive sequences called subterminal repeats (STRs), which are also called X element combinatorial repeats, and an X-core sequence.

The X-core sequence, localized on the centromere proximal side of the STRs, contains an autonomously replicating sequence (ARS) and binding sites for the transcription factor Abf1 [13]. Sir-proteins are recruited to the X-core sequences through interaction with the origin recognition complex (ORC) and Abf1. A few yeast telomeres lack the STR sequences, but all telomeres contain the X-core sequence.

The STRs are further subdivided (A, B, C and D) [14], where mostly STR-C and D contain several binding sites for the transcription factors Reb1 and Tbf1. Genome-wide chromatin immunoprecipitation using exonuclease I (ChIP-exo) showed that many, though not all, of the Reb1 binding sites in the STR loci are indeed occupied by Reb1 in vivo [15]. Both Reb1 and Tbf1 are essential proteins and Reb1 has been found to be a so-called pioneer transcription factor [16] that is able to bind its target sites even in condensed chromatin. It was suggested that Reb1 and Tbf1 have a role in limiting the spread of telomere silencing [17,18].

In yeast, telomere position effect (TPE) was first demonstrated using *URA3* or *ADE2* reporter genes juxtaposed to the ends of synthetic telomeres, resulting in reduced expression

of the reporters [7,19]. Like silencing of *HML*α and *HMR***a**, TPE depended on Sir proteins, although Sir1 had no effect [20]. The model proposed posited that a repressive chromatin structure spreads from the Rap1-bound telomeric repeats and extends inwards for a few thousand base pairs [19,21,22].

However, this model is oversimplified since insertions of *URA3* close to several natural telomeres showed that only 6 of the 17 telomeres tested displayed reduced *URA3* expression [23]. In addition, an RNA-sequencing study revealed that only 6% of subtelomeric genes increased expression in *sir* mutant strains [24]. Studies using ChIP-Seq of Sir proteins [25,26] demonstrated that Sir proteins associate at discrete positions at natural subtelomeres and that the highest level of enrichment was overlapping the X-core sequences. Consistently, the X-elements display Sir-dependent histone H4K16 hypoacetylation [25]. A model of TPE at natural telomeres suggests that silencing at telomeres is discontinuous, where the STRs can stop spreading of silent chromatin and the X-core sequences act as silencer elements together with the Rap1-bound terminal telomeric repeats [18].

Further evidence for the dynamic nature of telomeric heterochromatin came with the discovery of telomeric repeat-containing RNAs (TERRAs) [27,28]. TERRAs are noncoding RNAs, transcribed by RNA polymerase II, originating in subtelomeric sequences and extending into the terminal repeats [28–30]. TERRAs are conserved from yeast to humans and form RNA-DNA hybrids, called R-loops, at the telomeric repeats of critically short telomeres. A study suggested that during replication of the telomeres, these R-loops interfere with the replication fork, causing stalling and DNA double strand breaks [31]. These breaks can promote alternative lengthening of the telomere (ALT) through a recombination mechanism, using a sister chromatid with a longer telomere as template. Hence, TERRA R-loops may stimulate telomere maintenance in cells that lack expression of the telomerase enzyme. Since most somatic human cells have insufficient telomerase activity, TERRA R-loops could be important determinants of replicative senescence.

In this study, we investigated how Sir2 affected nucleosome positioning and occupancy using Micrococcal nuclease digestion followed by deep sequencing (MNase-Seq) combined with Micrococcal nuclease digestion followed by quantitative PCR (MNase-qPCR). Loss of Sir2 resulted in a remarkably consistent absence of nucleosomes in the STR-D regions of the X-elements at multiple telomeres. Reb1 loss-of-function conditions reestablished the "missing" nucleosome, suggesting a model in which Sir2 and Reb1 compete for stabilizing/destabilizing these nucleosomes. In addition, both Reb1 and Sir2 repressed TERRAs, apparently by distinct mechanisms.

## Results

### MNase-Seq of *sir2* mutants in *S. cerevisiae*

To profile the position and occupancy of nucleosomes in yeast cells lacking Sir2, we performed micrococcal nuclease digestion of chromatin and sequenced the mono-nucleosomal fraction using high throughput sequencing (MNase-Seq). The strains used were isogenic *MAT***a** (*WT*) and *MAT***a** *sir2*. After the reads were trimmed, normalized to 1 x input, aligned to the *S. cerevisiae* genome (Release 64), and analyzed using the DANPOS software [32], the results were visualized in the Integrated Genome Viewer (IGV) (Fig 1, [33]). DANPOS listed all nucleosome peaks in the genome (~70000 peaks in total). Next, the software calculated a difference value for the summit of the peaks comparing the two strains. DANPOS also calculated a false discovery rate (FDR) determining whether the difference between strains was statistically significant or not. We thus obtained a rank of the nucleosomes showing the largest difference between *sir2* and *WT* (S1 Table).

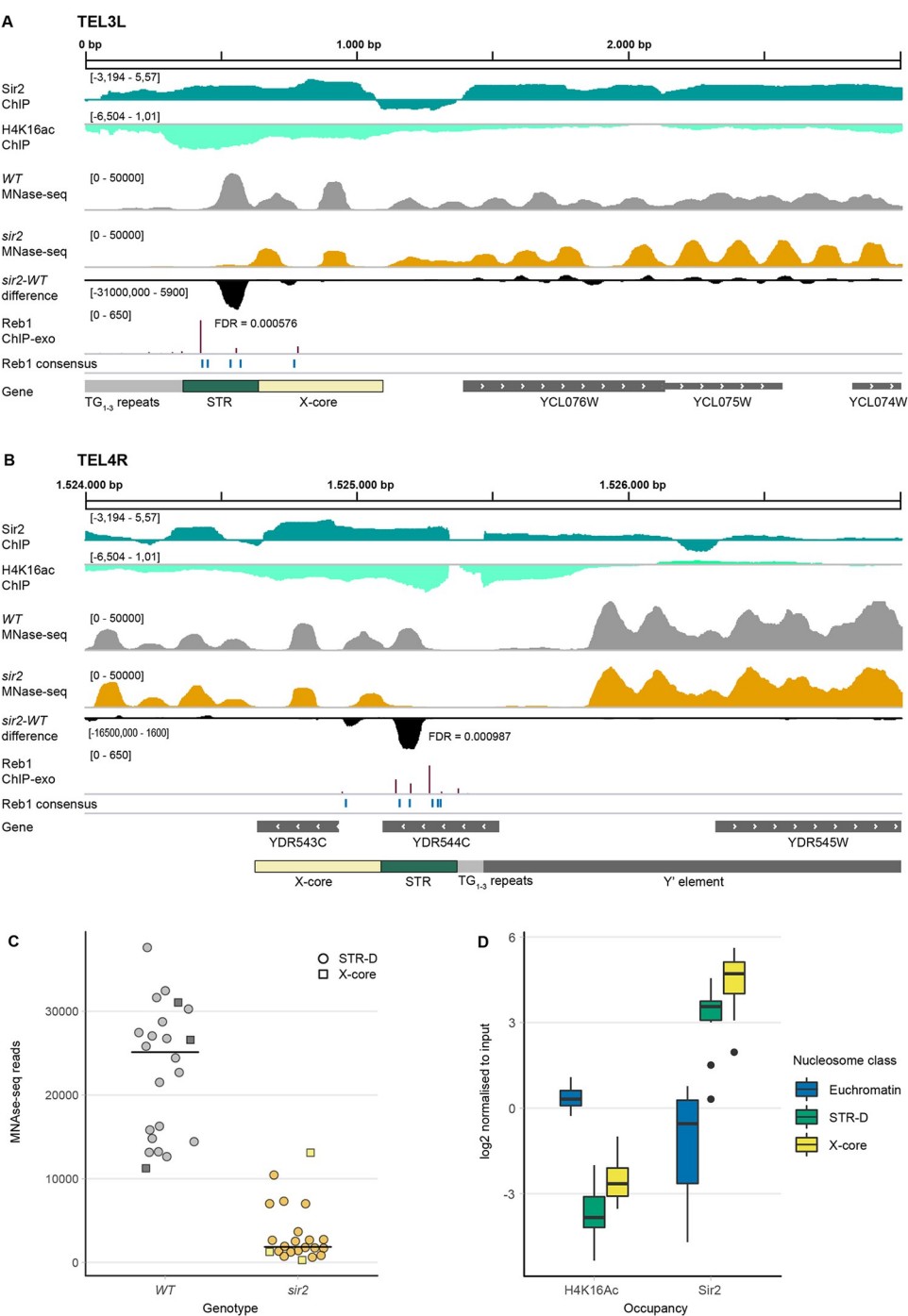

**Fig 1. Chromatin features of *WT* and *sir2* strains at the telomeres.** (**A**) Illustration of 3000 bp from the left telomere of chromosome 3 in the Integrative Genome Viewer (IGV). Shown are tracks generated from ChIP-seq data for Sir2 (dark green) and H4K16 acetylation (light green), followed by MNase-seq tracks of *WT* (grey) and *sir2* (orange) strains and the DANPOS-calculated difference between *sir2* and *WT* (black). FDR value is given under/adjacent to the nucleosome with the largest difference. Reb1 consensus sites are denoted in blue and Reb1 occupancy levels (ChIP-exo) are shown in dark red. Genes are shown in the bottom track in grey. (**B**) Same as in (**A**), except showing a 3000 bp region from the right arm of chromosome 4. Note that YDR544C is a dubious ORF, unlikely to encode a protein. (**C**) MNase-seq read counts of the summit of the nucleosomes covering the STR-D (circles) or X-core loci (squares) of 22 telomeres in *WT* (grey) and *sir2* (orange) strains. (**D**) Quantification of both acetylated H4K16 and Sir2 occupancy at the STR-D locus compared to the X-core region and euchromatin presented as box-plots. 15 STR-D loci were compared to 15 X-core loci and 24 randomly selected euchromatic nucleosomes.

In these strains the cryptic mating type loci remained intact, so in the *sir2* mutant strain we observed the indirect effects of the α2/Mcm1 and **a**1/α2 transcriptional repressor complexes (due to the derepression of *HMLα*), as well as direct effects at loci where Sir2 binds. In this study, we will focus on nucleosome differences observed in the subtelomeric loci. Nucleosome differences in the regulatory regions of cell-type specific genes and in the rDNA will be presented elsewhere.

## A nucleosome occupancy difference in the subtelomeric X-elements in *sir2*

Previous lower resolution investigations of nucleosome positioning in subtelomeric regions [23,34] indicated low nucleosome occupancy at the STR regions of X-elements, but high occupancy at Y′-elements. We obtained similar results, but observed a nucleosome occupying the centromere-proximal end of the STR sequences. Others have previously divided the STR sequence into A, B, C and D elements based on homology between telomere ends [14] and whereas the STR-A to -C elements had low nucleosome occupancy, there was a positioned nucleosome covering the STR-D elements at many telomeres. However, in the *sir2* strain the occupancy of the STR-D nucleosome was much lower than in the *WT*. Fig 1A and 1B show two examples, TEL3L (X-only) and TEL4R (containing a Y′ element). S1–S6 Figs display the nucleosome occupancy at 24 additional telomeres including FDRs of the nucleosome occupancy differences. The low occupancy nucleosome in the *sir2* strain corresponded to the STR-D elements at 17 telomeres and the nucleosome occupancy differences were highly significant. For six telomeres, the STR-D element was absent or truncated by homology searches (TEL5L, 6R, 8L, 9L, 10L and 16R), and three of those telomeres (5L, 8L and 16R) did not display significant nucleosome occupancy differences in *sir2*. Two telomeres having truncated STR-D elements (TEL9L and 10L) displayed a nucleosome positioning difference in the X-core element. For two telomeres (TEL6R and 8R) a significantly lower nucleosome occupancy corresponded to the X-core element.

For the remaining six telomeres, there were very few or no sequencing reads in the X-elements even in the *WT* (TEL1R, 3R, 4L, 7L, 13R and 14R). We suspected that there was an issue with the alignment to the R64-1-1 assembly, which is based on strain S288C, since the reads were obtained from the W303 strain background. To test this idea, we aligned the MNase-Seq reads to the whole-genome sequence of W303 [35]. For TEL1R, 3R, 7L and 13R, aligning to the W303 genome revealed a significant lowered nucleosome occupancy at the STR-D loci in the *sir2* strain (S7 and S8 Figs). TEL14R had a truncated STR-D element. Since the annotation of the W303 assembly was of lower quality compared to the R64-1-1assembly and genome coordinates changed between strains, we nevertheless preferred to use the R64-1-1 assembly for the following analyses.

Fig 1C displays the read counts of the summits of the nucleosomes covering the STR-D loci or the X-core (TEL6R, 8R and 10L) comparing 22 telomeres in the *WT* and *sir2* strains. In conclusion, Sir2 stabilized a nucleosome in the subtelomeric X-elements mostly corresponding to the STR-D element.

## The STR-D nucleosome had high Sir2 occupancy and low histone H4 lysine 16 acetylation

Others have previously performed ChIP-Seq on MNase-treated chromatin to investigate genome-wide Sir2 occupancy and histone H4K16 acetylation [25]. Using these data, we mapped Sir2 and histone H4K16ac at the STR region of telomeres. In Fig 1A and 1B, the top two tracks display these data, revealing high Sir2 occupancy and low histone H4K16 acetylation at the STR-D loci of chromosome 3L and 4R. We quantified Sir2 and H4K16 acetylation

at the STR-D loci at 15 telomeres, comparing to 24 randomly selected euchromatic loci. We also included the second centromere proximal nucleosome in relation to STR-D, roughly corresponding to the X-core sequences (Fig 1D). The results confirmed that high Sir2 occupancy and low histone H4K16 acetylation was a general feature of the STR-D and X-core loci.

## MNase-qPCR confirmed the MNase-Seq data

The STR-D-elements from different subtelomeres share ~80 to 90% sequence identity, so it was possible to assign them to a specific telomere using the ~150 bp paired-end reads obtained from the MNase-Seq experiment, but more difficult to differentiate them by the polymerase chain reaction (PCR). We used the existing polymorphisms to generate primer pairs 100% complementary to a single telomere, although in some cases, these primer pairs probably amplified STR-D elements from more than one telomere. Next, qPCR assays of MNase-treated genomic DNA were used to explore nucleosome occupancy (Fig 2A). To account for variations between samples, we normalized to a nucleosome in the *ACT1* gene, which was unchanged in *WT* and *sir2* in the MNase-Seq experiment. For 15 primer pairs used, we observed a 5- to 8-fold lower nucleosome occupancy in *sir2*, confirming the MNase-Seq results. Reassuringly, the primer pair for TEL5R displaying a less significant (FDR = 0.0012) difference in nucleosome occupancy between *WT* and *sir2* in the MNase-Seq reads also showed a smaller difference in the MNase-qPCR assay.

To exclude the possibility that the MNase-resistant DNA at the STR-D loci would consist of protein complexes distinct from a nucleosome, we immunoprecipitated MNase-treated genomic DNA using a histone H3 antibody. Quantitative PCR using primers specific to the STR-D loci at TEL3L, 4R and 11L showed that the DNA resistant to MNase digestion was efficiently immunoprecipitated by the histone H3 antibody (Fig 2B) from the *WT*, but not from the *sir2* strain. Hence, the MNase-resistant DNA at the STR-D loci were nucleosomes.

Because *sir2* strains derepress the cryptic mating type loci, a possibility was that the observed decreased nucleosome occupancy at the STR-D loci was due to an indirect effect on cell-type. This notion was unlikely as Sir2 displayed direct binding to the STR-D loci. We nevertheless tested this idea by exploring a strain containing deletions of the mating-type loci and *SIR2*, hence avoiding effects on cell-type. MNase-qPCR revealed that the absence of Sir2 and not cell-type caused the decreased nucleosome occupancy (S9A Fig).

At the cryptic mating-type loci and telomeres, Sir2 acts in a complex with Sir3/Sir4 and all three proteins are critical for maintaining a repressive chromatin structure. MNase-qPCR using a *sir3* mutant strain phenocopied the decreased nucleosome occupancy observed in the *sir2* strain. Hence, both Sir2 and Sir3 were required for maintaining the STR-D nucleosome (S9B Fig).

The histone deacetylase activity of Sir2 has been shown to critically depend on Asparagine 345 [36]. To test if the catalytic activity of Sir2 was important for the nucleosome occupancy phenotype, we examined a *sir2N345A* mutant strain by MNase-qPCR (Fig 2C). The results showed a reduction of nucleosome occupancy at the STR-D loci at TEL3L, 4R and 11L, though not to the same extent as the strain completely lacking Sir2. This suggested that the catalytic activity of Sir2 was important, but not essential for stabilizing the STR-D nucleosome (see discussion). Consistent with this notion, *hhf1,2K16R* (mimicking H4K16) and *hhf1,2K16Q* (mimicking H4K16-acetyl) mutant strains did not display a significant change in occupancy of the STR-D nucleosome (S9C Fig).

## Reb1 binding to the STRs

The STRs contain binding sites for the essential transcription factor Reb1. The genome-wide binding sites for Reb1 were previously determined using a Chromatin immunoprecipitation

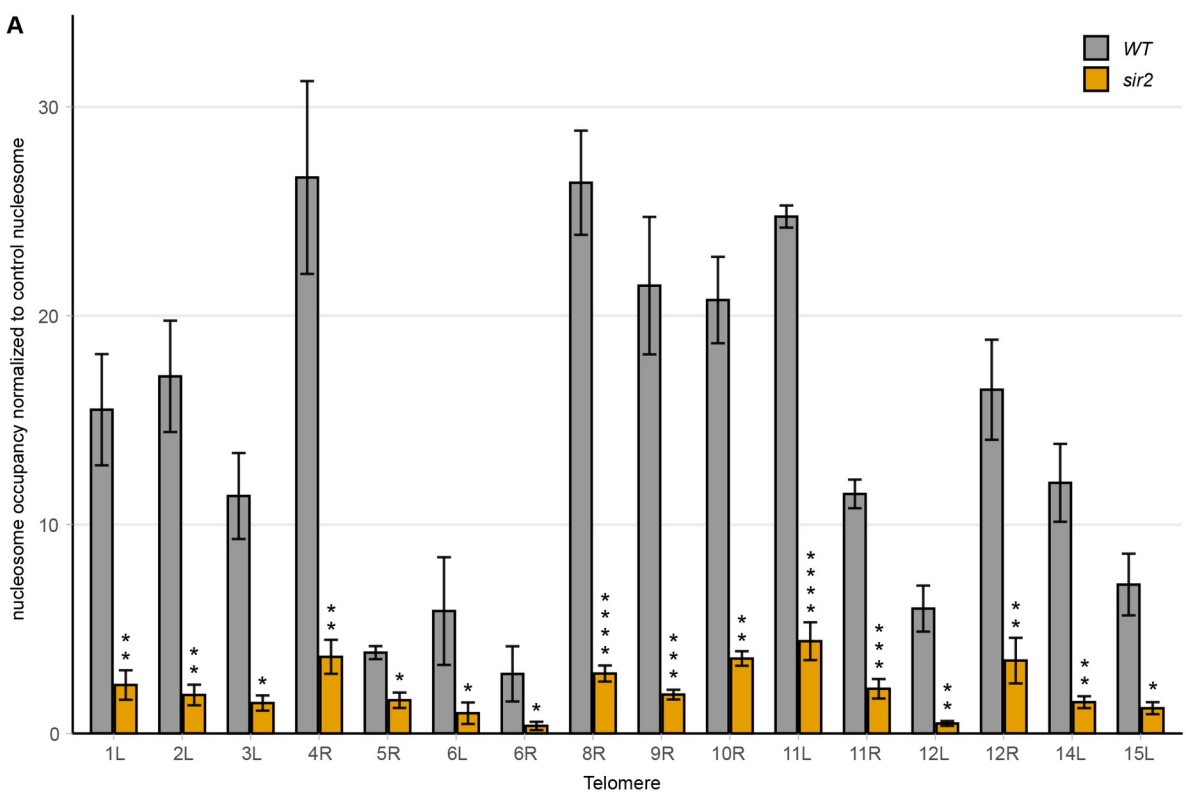

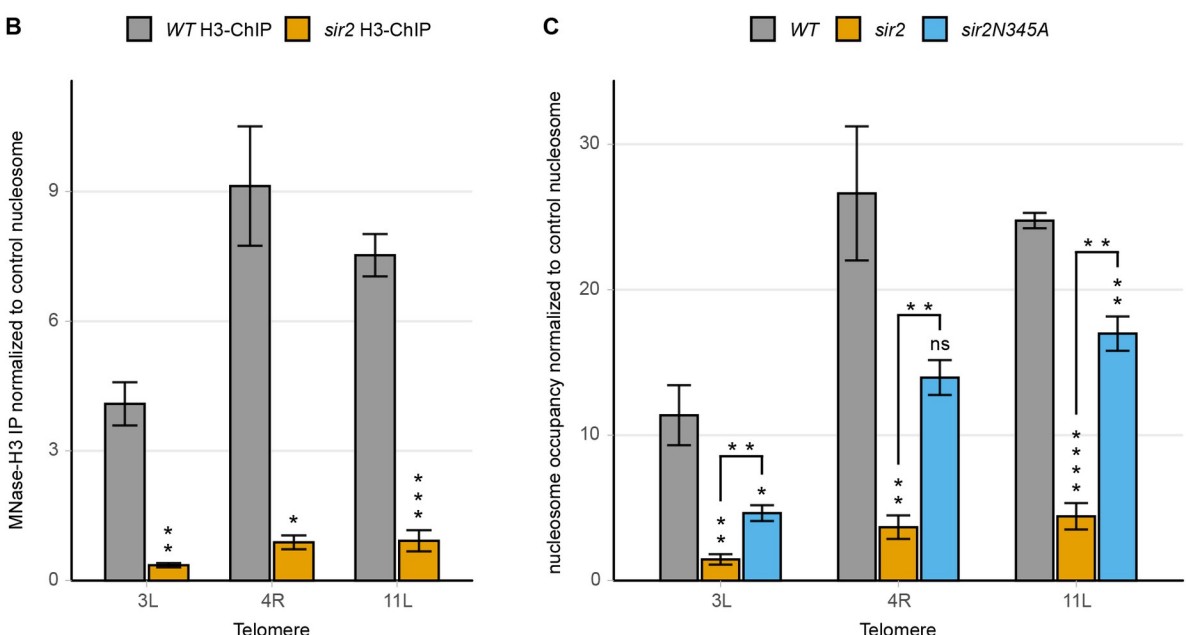

**Fig 2. Sir2 stabilizes a nucleosome in the X-elements.** (**A**) MNase-qPCR of the STR-D region of 16 telomere ends in *WT* and *sir2* strains. (**B**) MNase-H3-ChIP-qPCR of *WT* and *sir2* strains at the STR-D loci of TEL3L, 4R and 11L. (**C**) MNase-qPCR of the indicated STR-D loci in *WT*, *sir2* and *sir2N345A* strains. All data are shown as mean ± SEM (n = 3, $^*$p<0.05, $^{**}$p<0.01, $^{***}$p<0.001, $^{****}$p<0.0001, ns = not significant).

coupled with exonuclease 1 digestion method (ChIP-exo) [37] and Reb1 was shown to bind the STR-elements. This method determines the binding sites with close to base pair resolution. The STR-elements contain five to eight consensus-binding sites for Reb1 (TTACCCT), although the ChIP-exo data indicated that only some of these binding sites were occupied. We plotted Reb1 occupancy and Reb1 consensus sites at TEL3L and 4R (Fig 1A and 1B, lower tracks). There was a pattern of higher Reb1 occupancy at the telomere-proximal border of the affected nucleosome and the Reb1 consensus sites at positions occupied by the affected nucleosome were either lowly occupied or not bound at all. To explore whether this was a general pattern, we plotted a meta-nucleosome map, displaying the region overlapping the affected nucleosomes showing Reb1 occupancy as a heat map for 16 telomeres (9 left and 7 right, Fig 3A). This analysis showed that the most highly occupied Reb1 sites (purple) were on the telomere proximal border of the STR-D nucleosome. Many consensus sites bound by the nucleosome were not occupied and there was a nucleosome-depleted zone that extended over the entire STR A-C region. We presumed that the Sir-complex normally stabilizes the STR-D nucleosome, which may prevent Reb1 binding to the consensus binding sites in the STR-D region of the telomere.

## Reb1 depletion rescued the nucleosome occupancy defect in *sir2*

We next set out to test the effect of depleting Reb1 on nucleosome occupancy. Since Reb1 is essential, a temperature-sensitive allele of Reb1 was generated (see materials and methods).

The resulting *reb1* allele was temperature sensitive at 37˚C (S10A Fig) and contained three point mutations changing Glutamate 369 to Glycine, Serine 389 to Proline, and Aspartate 510 to Asparagine. The latter was situated in one of the two Myb-like DNA binding domains [38]. The steady-state level of mutant Reb1 did not change at the restrictive temperature (S10B Fig), suggesting that the temperature sensitivity was due to impaired DNA binding.

MNase-qPCR of the *reb1$^{ts}$* strain grown at the restrictive temperature showed that the occupancy of the STR-D nucleosome at TEL3L, 4R and 11L increased modestly (but significantly) compared to *WT*. Interestingly, Reb1 depletion rescued the nucleosome occupancy phenotype in the *sir2* strain (Fig 3B). Moreover, overexpression of Reb1 in a *sir2* strain further decreased nucleosome occupancy at the STR-D loci (S11A Fig). These observations were consistent with a model where Sir2 and Reb1 compete for binding at the STR-D loci, thus stabilizing/destabilizing the nucleosomes, respectively.

A prediction of this model was that more Reb1 should bind to the STR-D loci in a *sir2* strain compared to the *WT*. We could demonstrate this by ChIP in a *REB1-TAP sir2* strain (Fig 3C) for TEL3L, but not for TEL4R or TEL11L. We attribute this to technical limitations due to the clusters of occupied Reb1 binding sites at these loci. The relative large size of fragments (~300 bp), despite extensive sonication, limited the resolution of the ChIP, making it challenging to detect increased binding.

## Reb1 and Sir2 repressed TERRAs by distinct mechanisms

Others previously showed that TERRAs originate in the subtelomeres, extending towards the terminal repeats [39], and that TERRA levels are upregulated in a *sir2* strain [29]. However, whether Reb1 affected TERRA levels has not been explored.

Measuring TERRA levels from TEL1L, 3L and 15L in the *reb1$^{ts}$* strain at the restrictive temperature revealed a considerable increase of TERRAs (>75-fold compared to *WT*) (Fig 4A and Table 1). We also confirmed that TERRA levels increased in the *sir2* strain. Reb1 depletion caused a 4- to 10-fold increased TERRA abundance compared to the levels observed in the *sir2* strain. When we depleted Reb1 in a *sir2* strain, we observed a synergistic increase in TERRA

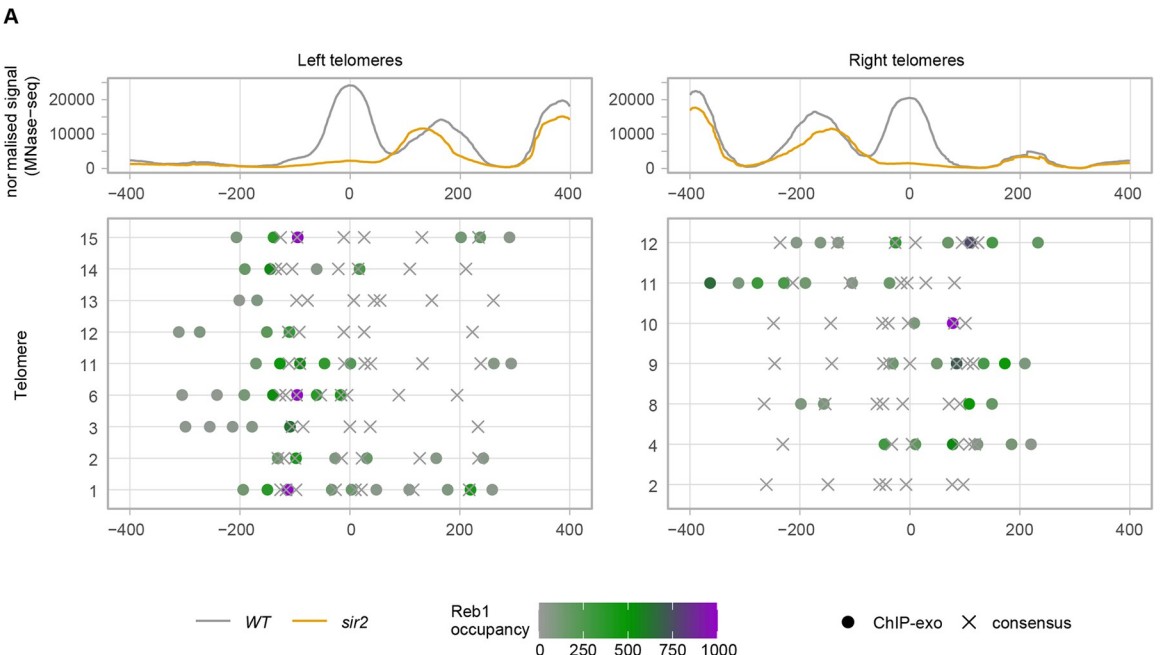

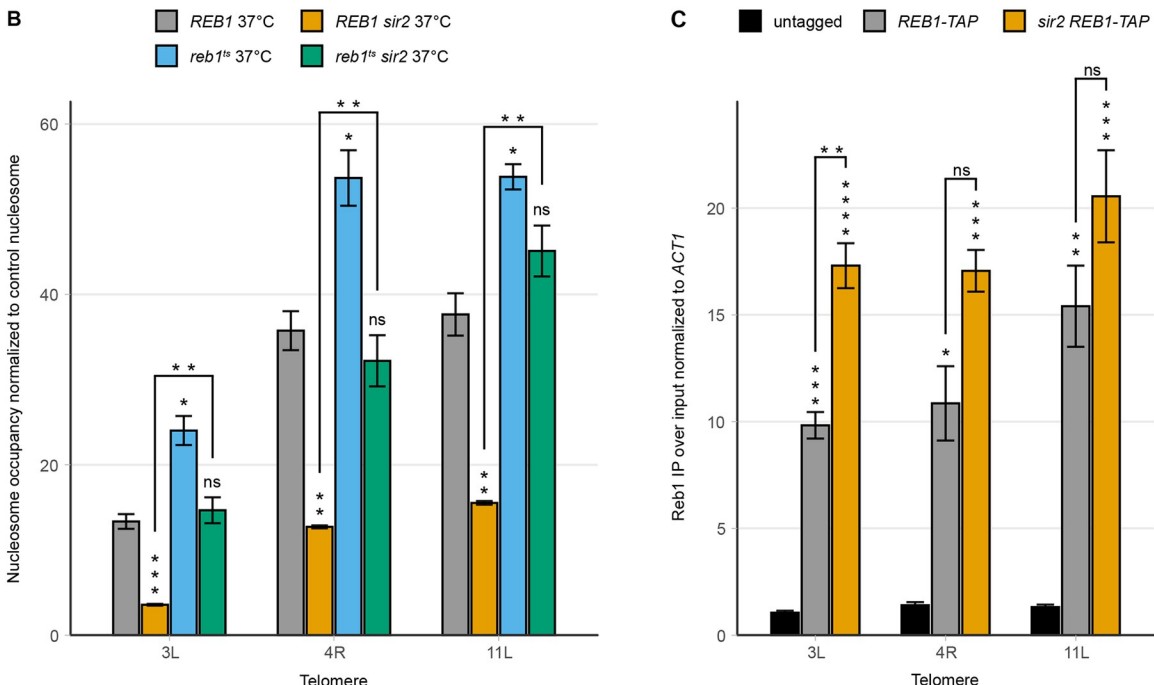

**Fig 3. Reb1 binds to subtelomeres and destabilizes nucleosomes.** (**A**) Meta-X-element analysis of 9 left and 7 right telomere ends. The top panel shows the normalized MNase-seq signal of *WT* (grey) and *sir2* (orange) centered on the summit of the nucleosome with the highest occupancy change. In the middle panel, Reb1 consensus sites (X) and occupancy (dots) presented as a heat map are shown for the individual X-elements. (**B**) MNase-qPCR of *WT* and *sir2* strains carrying a deletion of the endogenous *REB1* gene and expressing either a *WT* or mutant temperature-sensitive plasmid-borne allele of *REB1*. Strains were grown at restrictive temperature. (**C**) Chromatin immunoprecipitation followed by qPCR of the indicated strains. Ct values were first normalized to *ACT1*, followed by normalization to input. All data are shown as mean ± SEM (n = 3, *p<0.05, **p<0.01, ***p<0.001, ****p<0.0001, ns = not significant).

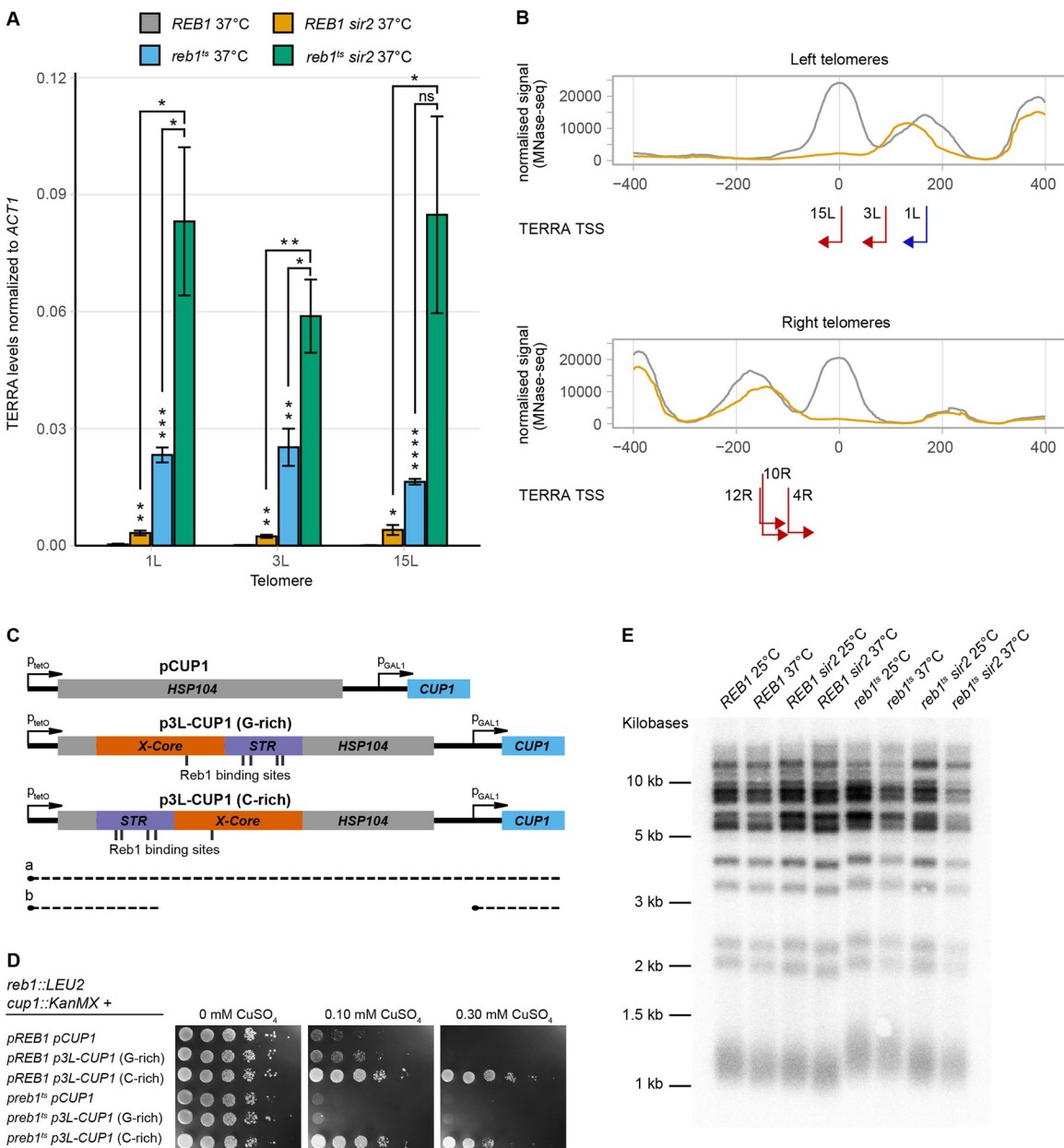

**Fig 4. Reb1 regulates TERRAs.** (**A**) TERRA RT-qPCR of *WT* and *sir2* strains dependent on either a *WT* or temperature-sensitive allele of *REB1* on a plasmid. Strains were grown at the restrictive temperature. All data are shown as mean ± SEM (n = 3, *p<0.05, **p<0.01, ***p<0.001, ****p<0.0001, ns = not significant). (**B**) Meta-X-element as in Fig 3A for left and right telomeres, with each TERRA TSS and their respective direction of transcription indicated by red arrows. Blue arrow depicts the TSS from TEL1L, determined previously. (**C**) Schematic for the Cu²⁺-resistance assay used to investigate transcriptional termination by the TEL3L X-element. Transcriptional read-through (a) will interfere with *CUP1* transcription and lead to Cu²⁺-sensitivity. Roadblock transcriptional termination (b) will enable *CUP1* transcription and confer resistance. (**D**) Cu²⁺-resistance assay. The indicated strains containing either empty vector (*pCUP1*) or the X-element (*p3L-CUP1*) in two different orientations spotted in ten-fold serial dilutions on SC plates lacking histidine and containing different concentrations of CuSO₄, as indicated. Cells were grown at 30˚C, which allowed normal growth of the *reb1^Ts* strain (left). (**E**) DNA blot analysis of telomere length for the indicated genotypes and temperatures. *Xho*I digested genomic DNA was probed with a telomeric repeat specific ³²P-end labeled oligonucleotide.

**Table 1. TERRA levels are regulated by Sir2 and Reb1.**

| Genotype | TERRA levels normalized to *ACT1* (SEM, n = 3) | fold increase over *REB1* 25°C | fold increase over *REB1* 37°C |
|---|---|---|---|
| **Telomere 1L** | | | |
| *REB1* 25°C | 0,00016 (0,0000152) | **1,0** | 0,5 |
| *REB1* 37°C | 0,00031 (0,0001734) | 1,9 | **1,0** |
| *REB1 sir2* 25°C | 0,00078 (0,0003680) | 4,8 | 2,5 |
| *REB1 sir2* 37°C | 0,00324 (0,0005957) | 20,2 | 10,5 |
| *reb1^{ts}* 25°C | 0,00028 (0,0000469) | 1,7 | 0,9 |
| *reb1^{ts}* 37°C | 0,02324 (0,0019165) | 145,2 | 75,0 |
| *reb1^{ts} sir2* 25°C | 0,00351 (0,0005994) | 21,9 | 11,3 |
| *reb1^{ts} sir2* 37°C | 0,08314 (0,0189999) | 519,4 | 268,5 |
| **Telomere 3L** | | | |
| *REB1* 25°C | 0,00014 (0,0000081) | **1,0** | 0,9 |
| *REB1* 37°C | 0,00015 (0,0000206) | 1,1 | **1,0** |
| *REB1 sir2* 25°C | 0,00066 (0,0001793) | 4,8 | 4,4 |
| *REB1 sir2* 37°C | 0,00239 (0,0003955) | 17,2 | 16,0 |
| *reb1^{ts}* 25°C | 0,00060 (0,0000814) | 4,3 | 4,0 |
| *reb1^{ts}* 37°C | 0,02522 (0,0047691) | 181,8 | 168,5 |
| *reb1^{ts} sir2* 25°C | 0,00393 (0,0007083) | 28,3 | 26,3 |
| *reb1^{ts} sir2* 37°C | 0,05886 (0,0093934) | 424,3 | 393,3 |
| **Telomere 15L** | | | |
| *REB1* 25°C | 0,00003 (0,0000009) | **1,0** | 0,5 |
| *REB1* 37°C | 0,00006 (0,0000098) | 2,1 | **1,0** |
| *REB1 sir2* 25°C | 0,00060 (0,0001151) | 19,7 | 9,3 |
| *REB1 sir2* 37°C | 0,00402 (0,0012802) | 132,2 | 62,4 |
| *reb1^{ts}* 25°C | 0,00047 (0,0000989) | 15,5 | 7,3 |
| *reb1^{ts}* 37°C | 0,01637 (0,0007170) | 539,0 | 254,4 |
| *reb1^{ts} sir2* 25°C | 0,00342 (0,0009561) | 112,4 | 53,1 |
| *reb1^{ts} sir2* 37°C | 0,08485 (0,0252368) | 2793,1 | 1318,6 |

steady-state levels (>270- fold compared to *WT*), indicating that Reb1 and Sir2 operate in separate pathways to regulate TERRA expression.

Reb1 can restrict cryptic and read-through transcription at internal loci via a so-called roadblock transcriptional termination mechanism [40,41]. We hypothesized that Reb1 may restrict TERRAs through a similar mechanism at telomeres. To investigate if subtelomeric X-elements could promote transcriptional termination, we used a plasmid-based assay in which the X-element from TEL3L was cloned in between a tetracycline-repressible promoter and a *GAL1* promoter. In the presence of a termination signal between the two promoters, the tetracycline promoter no longer interferes with the *GAL1* promoter [42]. The *GAL1* promoter drives the expression of a gene mediating copper resistance (*CUP1*), making growth in the presence of $Cu^{2+}$ the readout of the assay. The TEL3L X-element cloned in both orientations (Fig 4C) mediated a higher $Cu^{2+}$ resistance compared to the control plasmid (Fig 4D, top three rows). If the telomere proximal end of the X-element was oriented closest to the tetracycline promoter, a higher $Cu^{2+}$ resistance was observed (see 0.1 mM plate), revealing a partial orientation dependence, which has been observed before with the internal Reb1 consensus site of TTACCCG [41]. Therefore, in this assay, the subtelomeric X-element mediated transcriptional termination.

We noticed increased TERRA levels in the *reb1^{ts}* strain at the permissive temperature (Table 1), indicating that mutant Reb1 partially lost function also at lower temperatures.

Taking advantage of this observation, we also performed the assay for transcriptional termination in the *reb1*[ts] strain, growing the cells at 30˚C. The control plate, lacking Cu$^{2+}$, revealed the *reb1*[ts] strain grew normally at this temperature. However, Cu$^{2+}$ resistance was lower in the *reb1*[ts] strain compared with the *REB1* strain (Fig 4D). Thus, both the X-element and a fully functional Reb1 were necessary for efficient transcriptional termination in this assay.

## The 5′ ends of TERRAs resided in the X-element

Others have previously mapped the 5′end of TERRAs at telomere 1L [39]. Interestingly, at TEL1L, the TERRA transcriptional start site (TSS) was located just centromere proximal to the Sir2-dependent nucleosome, making this the +1-nucleosome relative to the TSS. To investigate TERRA TSS at other telomeres, we performed RNA-Mediated 5′ Rapid Amplification of cDNA Ends (RLM—5′RACE) using RNA that had been thoroughly treated to remove any residual DNA contamination (see materials and methods). We amplified TERRAs with different primers residing in the telomere proximal part of X-elements, expecting to enrich for TERRAs from X-only telomeres. S11B Fig is an example of nested PCR amplifications of 5′RACE using primers specific for TEL6R. After cloning and sequencing of amplified fragments, the results were subjected to BLAST-searches. By using RNA from both *WT* and *sir2* strains, we identified the 5′ ends of TERRAs originating from TEL3L, 4R, 10R, 12R, and 15L. The results were the same from the *WT* and *sir2* strains; hence, Sir2 did not influence TERRA TSS. Given the high similarity between X-elements from different telomeres, we only considered alignments that were > 95% identical. The results showed that the TERRA TSS map to sites just centromere-proximal to the STR-D nucleosome, consistent with the previous study [39]. The distance between the summit of the STR-D nucleosome (as defined by the MNase-Seq experiment) and the TERRA TSS were 45 bp (3L), 48 bp (4R), 73 bp (10R), 76 bp (12R) and 2 bp (15L) (Fig 4B). Hence, for these telomeres, the nucleosome stabilized by Sir2 was the +1 nucleosome for TERRA transcription. It should be noted that our analyses were biased for TERRAs from X-only telomeres and that cryptic promoters may also reside in Y′-elements.

## Lack of correlation between TERRA expression and telomere length

Previous studies have linked TERRAs with telomere length, demonstrating that high TERRA levels correlate with shorter telomeres [28,29], but also that TERRAs can act as a scaffold for telomerase components at critically short telomeres [43]. There is not a direct correlation between TERRA levels and telomere length however, since some mutants displaying high TERRA levels have long telomeres and vice versa [29]. Given the wide variation in TERRA levels found in the *sir2*, *reb1*[ts] and *sir2 reb1*[ts] strains, we investigated telomere length in these strains at both the permissive and restrictive temperatures (Fig 4E). The *reb1*[ts] strain had approximately 100 bp longer telomeres compared to the *WT*, irrespective of temperature. In addition, the *sir2 reb1*[ts] double mutant had longer telomeres than *WT*, but slightly shorter than the *reb1*[ts] single mutant. Since the *sir2 reb1*[ts] double mutant had much higher TERRA levels at the restrictive temperature compared to the *WT* strain (Table 1), we concluded that there was no correlation between TERRA levels and telomere length in these strains. However, a fully functional *REB1* gene limited telomere length.

## Proximity to the telomere was important to maintain the nucleosome positioning/occupancy in the TEL3L X-element

Next, we investigated whether the nucleosome occupancy of the X-element was dependent on the proximity to the telomere or not. We generated a strain in which the X-element at TEL3L was deleted from the subtelomere and reinserted at the euchromatic *URA3* locus. We then

designed primer pairs spanning the entire TEL3L X-element, taking advantage of as many polymorphisms as possible compared to other X-elements (S3 Table and Fig 5A), while also optimizing annealing temperatures to avoid unspecific amplifications. The TEL3L X-element deletion strain was used as negative control to validate the specificity of the primer pairs (Fig 5).

At the euchromatic position, both nucleosome occupancy and positioning were different compared to the native position at the telomere (Fig 5B). The previously observed reduction of nucleosome occupancy in a *sir2* strain was not detectable at the euchromatic position. In fact, we observed an increased nucleosome occupancy at the STR-D sequence in the *sir2* strain when the X-element was moved internally (Fig 5).

Additionally, the positioning of the nucleosomes in the artificial construct differed from the positions at the native TEL3L locus. The tiling MNase-qPCR assay suggested increased nucleosome occupancy of the STR-element (amplicon 1) and a positional change of the nucleosome covering the X-core region (amplicon 5).

We concluded that for the TEL3L X-element, the position close to the telomere, but not the sequence of the region itself, played an important role for both nucleosome position and occupancy.

## Discussion

In a large effort investigating chromatin interaction of roughly 400 different proteins in yeast, Pugh and co-workers found that the subtelomeric X-core element contained a stable triple nucleosome ensemble, which was bound by Sir-proteins [44]. The work presented herein agrees well with this observation and shows that this triple nucleosome structure depends on Sir2. In the absence of Sir2, the telomere-proximal nucleosome of this ensemble is destabilized and partially lost. By moving an X-element into euchromatin, we showed that both the triple nucleosome ensemble and the Sir2-stabilization was a position effect rather than a sequence-specific effect. Hence, the proximity to the telomere was critical to maintain this structure even in the presence of a full set of Sir-proteins. The 32 *S. cerevisiae* telomeres cluster into three to six foci at the nuclear envelope [45]. Mutations that disrupt the localization at the nuclear periphery exhibit partial derepression of telomere silencing [46] and altered recombination in subtelomeres [47,48]. A high concentration of silencing factors at the nuclear periphery probably contributes to these observations. The position effect of the nucleosomes of the X-element may also depend on the proximity to the terminal telomeric repeats, which recruit Sir-proteins independently of the X-core sequence. However, telomeres containing Y´-elements in which the terminal repeats are > 6 kb distant from the X-elements also depended on Sir2 for stabilization of the STR-D nucleosome. Telomere anchoring-pathways are redundant [49], so in order to study the relationship between telomere anchoring to the periphery and the nucleosome positioning/occupancy at X-elements, it will be necessary to perform a thorough analysis of such pathways.

The rDNA enhancer binding protein 1 (Reb1), first characterized over 30 years ago [50], is an abundant (~5000 molecules/cell) transcription factor regulating both RNA polymerase I and II genes and Reb1 binding sites are enriched for nucleosome free regions [51]. Reb1 represses cryptic transcripts through inhibition of transcription initiation [52], but also restricts cryptic and read-through transcripts by a roadblock transcriptional termination mechanism [41,53]. In the plasmid-based assay used in this study, we could demonstrate that the TEL3L X-element indeed could terminate transcription in a Reb1-dependent manner. This is consistent with the observation that TERRA steady-state levels increased after Reb1 depletion. Moreover, mapping TERRA TSSs showed that most of the Reb1 binding sites in the

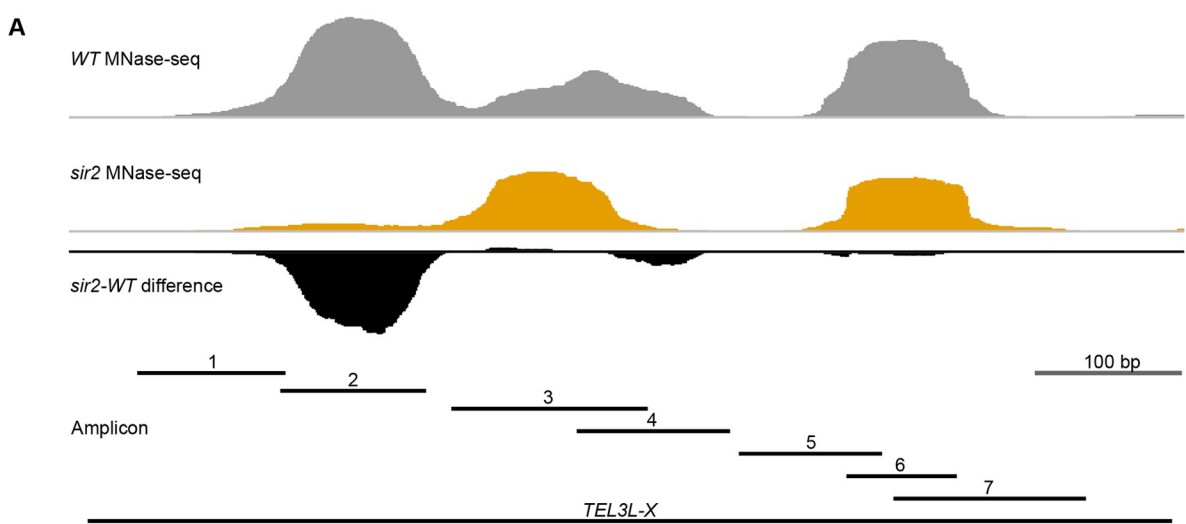

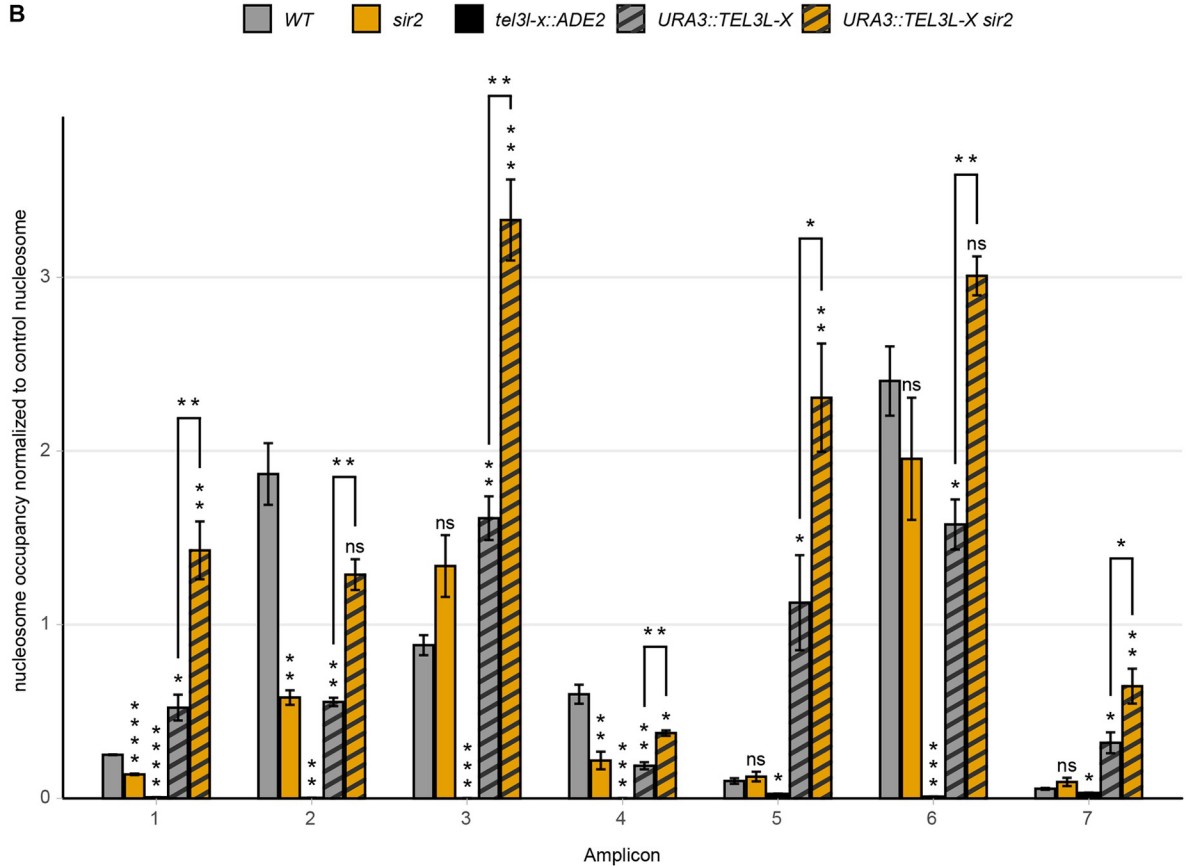

**Fig 5. The TEL3L X-element displays different nucleosome positioning/occupancy in euchromatin compared to its native location.**
(**A**) Schematic of amplicon location in the X-element. Top traces are MNase-seq signal of *WT* (grey), *sir2* (orange) strains, and the *sir2-WT* difference (black) at the native subtelomeric location. Bottom line indicates the stretch of DNA used for euchromatic insertion (*TEL3L-X*) and shorter, numbered lines indicate the amplicons. (**B**) Tiling MNase-qPCR of the indicated strains and amplicons along the TEL3L X-element. All data are shown as mean ± SEM (n = 3, *p<0.05, **p<0.01, ***p<0.001, ****p<0.0001, ns = not significant).

X-elements resided downstream from the TSS, suggesting that Reb1 restricts TERRAs by a roadblock transcriptional termination mechanism. However, we cannot exclude that Reb1 also affects transcription initiation of TERRAs. CRosslinking-Analysis of cDNA-sequencing (CRAC-Seq) data, which detects the position of RNA polymerase II by sequencing the nascent RNA, was performed in cells depleted for Reb1 [40]. However, these data are inconclusive with respect to TERRA transcription as they detect very little nascent transcripts in subtelomeric X-elements.

We suggest a competition model in which Sir2 normally stabilizes the triple nucleosome structure at X-elements by occluding Reb1 from binding its consensus sites in the STR-D loci (S12A–S12C Fig). Three observations are consistent with this model. (1) Lower Reb1 binding to the sequence occupied by the STR-D meta-nucleosome compared to the Reb1 consensus sites in the telomere proximal flanking sequences. (2) In a strain lacking Sir2, Reb1 depletion reinstated the triple nucleosome ensemble and (3) overexpression of Reb1 exacerbates the lowered nucleosome occupancy of the STR-D nucleosome in the absence of Sir2. We also observed a significant increased binding of Reb1-TAP to the TEL3L X-element in a *sir2* mutant strain, but it remains to be investigated whether more Sir-proteins bind to X-elements when Reb1 is inactivated.

The more than additive effect of removing Sir2 and depleting Reb1 on TERRA levels may be due to both increased transcriptional initiation and decreased transcriptional termination. The mapping of TERRA TSSs indicated that the Sir-stabilized nucleosome was the +1 nucleosome for TERRA transcription. The meta-nucleosome map of the X-elements also indicated that the middle nucleosome of the triple nucleosome ensemble subtly changed position in the *sir2* strain. Since nucleosomes are general inhibitors of transcription [54,55], it is reasonable to assume that the increased TERRA levels observed in the absence of Sir2 are the result of increased transcription initiation. However, lowered nucleosome occupancy of the STR-D nucleosome did not perfectly correlate with increased TERRA levels as the above-mentioned *sir2 reb1^{ts}* strain had very high TERRA levels and also a restored STR-D nucleosome.

The *sir2N345* mutant strain is both sterile and defective for TPE [36] so transcriptional silencing depends critically on the catalytic activity of Sir2. Sir3 displays reduced recruitment to TEL6R in the *sir2N345* background [36], and silencing models posit that Sir2-mediated deacetylation of histones promotes spreading of silent chromatin. However, the *sir2N345A* mutant had a weaker phenotype with respect to nucleosome occupancy in the X-elements compared to the complete lack of Sir2. In addition, a strain in which the only copies of Histone H4 were a histone H4K16Q mutant, mimicking acetylated H4K16, did not have a significant change of the nucleosome occupancy at the X-elements. Sir2 deacetylates other lysine residues than H4K16 (and possibly other unknown proteins), which may explain the lack of effect in the H4K16Q strain. The Sir2N345A result suggests that the role of Sir2 to deacetylate X-element nucleosomes was important, but not essential to stabilize the triple nucleosome ensemble. This result was not entirely surprising, as the Sir2N345A protein is recruited to the nucleation point (the silencers) at the cryptic mating type loci [25] and some recruitment of the Sir-complex was seen at positions internal to the silencers in the *sir2N345A* strain, indicating a limited spreading of silent chromatin. As the STR-D loci are close to the X-core, where the Sir-complex is recruited, Sir2N345A may still have a weak stabilizing effect on the STR-D nucleosome.

It has been suggested that TERRA-dependent R-loops at telomeres [56] result in replication fork collapse [57] and activation of a DNA damage response, promoting telomere instability [31]. In cells lacking telomerase expression, TERRA-dependent R-loops might facilitate the recruitment of homologous recombination proteins, thus promoting alternative telomere lengthening (ALT) [58,59]. Since a subset of cancers use ALT for telomere elongation,

manipulating TERRA expression is a possible therapy [60]. The role of telomere chromatin structure, telomere anchoring to the nuclear periphery, binding of Myb-domain proteins to subtelomeres and regulation of TERRA expression will be an important focus of continued research.

## Materials and methods

### Yeast strains and plasmids

The yeast strains used in this study are listed in S2 Table, oligonucleotides in S3 Table and plasmids in S4 Table. Yeast transformations were performed using the LiAc method [61], verified by PCR, test-mating, western blotting and/or sequencing. Gene deletions were, unless otherwise noted, generated by one-step gene disruptions using PCR fragments amplified from plasmids pFA6a-*KanMX* or pAG25 (*NAT*) [62,63] with 50 bp homology to the genomic locus or PCR fragments amplified from yeast strains containing the deletions. Bacterial transformations were performed with the CaCl$_2$-method into *E. coli* DH5α.

Plasmid pBM272-41 with P$_{GAL1}$-*REB1*, *HIS3* was isolated from J342 [38,50] and used for transformation of SAY75 and SAY2185. For the Reb1 overexpression experiment, transformed cells were grown in SD media plus appropriate amino acids and 2% glucose to saturation, diluted to an OD$_{600}$ = 0.1 in SD media plus 2% glucose or galactose and grown for 6 hours before being processed for MNase-qPCR.

Plasmid pST112 was generated by amplifying the *REB1* locus plus 568 bp upstream and 342 bp downstream of the ORF with primers adding restriction sites for *Xho*I and *Sac*I to the fragment. The resulting product was digested with aforementioned enzymes and ligated into pRS416 [64].

Plasmid pTG01 was constructed by PCR-amplifying the entire 738 bp X-element from TEL3L (*TEL3L-X*) using primers adding a *Bam*HI restriction site at the centromere-proximal part of the fragment and a *Sac*I restriction site at the telomere-proximal part of the fragment, digested and cloned into the corresponding sites of pRS406 [64]. The plasmid was verified by sequencing (Eurofins Genomics, Ebersberg, Germany). pTG02 was constructed similarly to pTG01, but using primers adding a *Sac*I restriction site at the centromere-proximal part of the fragment and a *Bam*HI restriction site at the telomere-proximal part of the fragment, thus reversing the orientation.

SAY2234 (*tel3l-xΔ::ADE2*) was generated by transforming SAY2208 with a ~2 kb *ADE2* PCR fragment (including 218 bp upstream of the start codon) having 50 bp homology to down- and upstream sequences of the *TEL03L-X* locus. SAY2234 has *ADE2* inserted with the promotor being centromere-proximal. Due to the low complexity of the X-elements, 41 bp of the telomere-proximal part of *TEL03L-X* were left intact, but the other 697 bp were replaced by *ADE2*.

SAY2301 and SAY2330 were constructed by transforming SAY2234 with ~0.1 μg of *Stu*I-digested pRS406 and pTG02, respectively.

D Libri kindly provided the pCM190-*HSP104-Gal1-CUP1* (hereafter-called p*CUP1*) plasmid [42]. The *CUP1* plasmids containing the *TEL3L* X-element were generated by digesting p*CUP1* in the partial *HSP104* ORF with *Pfl*MI and *Tth*111I (New England BioLabs, Ipswich, Massachusetts, USA) and purified by gel extraction (QIAEX II Gel Extraction Kit, Qiagen Sciences, Germantown, Maryland, USA). *TEL3L-X* was PCR-amplified from pTG01 with primers adding a 60 bp overhang homologous to *HSP104* in both ends. The plasmids containing either the G-rich (pTG04) or C-rich (pTG05) orientation of the Reb1 binding sites in the *TEL3L-X* insert were constructed by homologous recombination in a *WT* yeast strain. The plasmids were rescued from yeast, amplified in *E. coli*, test-digested and verified by sequencing

(Eurofins Genomics). The p*CUP1*, pTG04 and pTG05 plasmids had the auxotrophic marker *URA3* exchanged for *HIS3*. To perform the marker change, the plasmids were digested with *Eco*RV and *Stu*I, removing part of the *URA3* gene. *HIS3*, including the promotor and terminator sequence, was PCR-amplified from pRS423 with primers adding 60 bp of homology to the p*CUP1* backbone up- and downstream of the *URA3* cassette. The marker changed plasmids (pTG06, pTG07 and pTG08, respectively) were constructed by homologous recombination in a *WT* yeast strain and processed as described above.

## Generation of the *reb1* temperature sensitive allele

Error-prone PCR was performed with modifications from [65]. In total, four PCR reactions were prepared, each containing 10 mM Tris-HCl pH 8.3, 50 mM KCl, 7 mM MgCl$_2$, 0.2 mM dATP, 0.2 mM dGTP, 2 µM forward primer (REB1-1F), 2 µM reverse primer (REB1-1R), 20 pg/µl template DNA (pST112) with 0.05 U/µl *Taq* DNA polymerase (NEB). The first reaction contained 0.2 mM dCTP and 0.2 mM dTTP, the second contained 1 mM dCTP and 0.2 mM dTTP, the third contained 0.2 mM dCTP and 1 mM dTTP and the fourth contained 1 mM dCTP and 1 mM dTTP. The PCR products were precipitated and digested with *Dpn*I.

Yeast transformation was performed with ~0.1 µg of *Eco*RI and *Hin*dIII double digested pST112 together with ~1 µg PCR products from each reaction into yeast strain SAY200 (*reb1*Δ::*LEU2* + pBM272-41) and transformants were selected for on SC plates lacking uracil + 2% glucose at 25˚C. Cleaving pST112 with *Eco*RI and *Hin*dIII removes a stretch of 593 bp between Glutamic acid 205 and Phenylalanine 405. The error-prone PCR used primers spanning most of the ORF but not the promotor or terminator sequence, to ensure we only introduced mutations in the ORF sequence. Thus, the forward primer annealed at the start codon and the reverse primer annealed 57 bp upstream of the stop codon. The resulting transformants were pooled, grown overnight in liquid SC -uracil + 2% glucose at 25˚C and plated on SC -uracil + 2% glucose with approximately 150 CFUs per plate and incubated at 25˚C for three days, replicated onto new SC -uracil + 2% glucose plates placed at 37˚C. Colonies having a growth defect at 37˚C were picked from the original plate, followed by plasmids rescue from yeast, amplication in *E. coli* and sequencing of the resulting plasmid candidates (Eurofins Genomics).

## Western blot

3 OD$_{600}$ units of cells were harvested by centrifugation, resuspended in 100 µl of 1.85 M NaOH containing 7% β-Mercaptoethanol and incubated on ice for 10 min, followed by addition of 100 µl of 50% TCA and incubation on ice for 5 min. After centrifugation at 13,000 g for 10 min, protein pellets were washed twice with 1 M Trizma base (Merck, Darmstadt, Germany) before resuspension in 100 µl 2× SDS-PAGE loading buffer (100 mM Tris-HCl pH 6.8, 2% SDS, 10% glycerol, 4 mM EDTA, 0.2% bromophenol blue, 2% β-Mercaptoethanol). Samples were incubated at 95˚C for 10 min, briefly centrifuged and analyzed by SDS-PAGE and immunoblotting using anti-Myc antisera (9E11, Santa Cruz Biotechnology, Dallas, Texas, USA). Immunoreactive signals were detected with a horseradish peroxidase-conjugated secondary antibody (P0447, Dako, Santa Clara, California, USA) reacted with ECL Select (GE Healthcare, Little Chalfont Buckinghamshire, United Kingdom) and images were acquired with a FujiFilm LAS-1000 camera using the Intelligent Dark Box II (FujiFilm, Tokyo, Japan).

## Southern blot

Genomic DNA was digested with *Xho*I, electrophoresed on a 1% agarose gel and transferred to a Biodyne B 0.45 µm membrane (Pall Life Sciences, Pensacola, Florida, USA) by capillary

action using 10x SSC overnight. DNA was cross-linked to the membrane using the UV Strata-linker 1800 (2 times 70000 μJ/cm$^2$). A telomeric 5′-CACCACACCCACACACCACACC-CACA-3′ oligo was 5′ labelled by incubating with T4 Polynucleotide Kinase (NEB) and ATP (γ-$^{32}$P) (Perkin-Elmer, Walton, Massachusetts, USA) according to manufacturer recommendations and used for hybridization. Membranes were pre-hybridized and hybridized for 75 min at 50˚C with Church buffer (0.25 M Sodium phosphate buffer pH = 7.2, 1 mM EDTA, 1% BSA, 7% SDS) and washed four times with 6x SSC + 0.1% SDS. The BAS-MS imaging plate (FujiFilm) was exposed to the membrane overnight and images were acquired with a FujiFilm FLA-3000 Phosphorimager.

## MNaseSeq experiments

Mononucleosomes were prepared as described [66]. Libraries were prepared using Thru-PLEX-seq (Rubicon Genomics, Ann Arbor, Michigan, USA) and sequenced using the HiSeq2500 system and v4 sequencing chemistry (Illumina, San Diego, California, USA). Following trimming off low quality bases, reads were mapped to reference genome R64-1-1 using bowtie 1.1.2 [67]. Reads with multiple best-score alignments ("multi-mapping reads") were reported as one randomly selected alignment to allow for analysis of repetitive regions while reducing the overrepresentation of individual homologous regions in read pileups. Resulting alignments were converted to bam format, sorted and indexed using SAMtools 1.3 [68]. DAN-POS was used for detection of nucleosome positions and occupancy [32]. Coverage tracks normalized to 1x coverage were computed using deepTools 2.3.1 [69].

## MNase-qPCR

Mononucleosomes were prepared following the protocol described in [66], with minor modifications. Cell pellets were resuspended in 400 μl BB (10 mM Tris-HCl pH 8, 1 mM CaCl$_2$, 0.5 mM EDTA). Glass bead lysis was performed at 4˚C four times 90 sec with 2 min on ice in between. 1 mg of total protein was digested in RB (10 mM Tris-HCl pH 8, 2 mM CaCl$_2$ and 20 units Micrococcal nuclease (NEB)) in a total reaction volume of 200 μl. MNase-digested samples were separated on 1.5% agarose gels and mononucleosome bands were excised. DNA was extracted using QIAEX II Gel Extraction kit (Qiagen), followed by purification using PCR Purification kit (Qiagen) and dilution to 0.1 ng/μl.

Quantitative PCR was performed using primers specific for the locus covered by the nucleosome in the X-element that showed occupancy change in the MNaseSeq data and normalized to a region in the *ACT1* gene where nucleosome occupancy did not change (S3 Table).

Quantitative PCR was performed using a C1000 Thermal Cycler equipped with the CFX96 Real-Time System (Bio-Rad, Hercules, CA, USA) and iQ SYBR Green Supermix (Bio-Rad). Annealing temperature was 55˚C except for the TEL3L tiling experiment, which was performed at 66˚C to optimize specificity of the primers.

## MNase-ChIP-qPCR

Mononucleosomes were prepared as described above. Approximately 6 mg of total protein were digested with an amount of MNase previously determined to result in mostly mono-nucleosomes in 600 μl RB for 1 hour at 37˚C.

The standard ChIP protocol [70] was adapted as follows: The lysate-reaction mixture was cleared by centrifugation at 14000 rpm and 4˚C for 4 min. The supernatant was transferred to a new tube and FA lysis buffer was added up to 1 ml. 20 μl equilibrated Dynabeads Protein G (Invitrogen, Waltham, Massachusetts, USA) were added to the lysate and incubated for 4 hours at 4˚C to pre-clear. Supernatant was transferred to a new tube and 6 μg H3-antibody

(Abcam 1791) were added. The mixture was incubated with rotation at 4˚C overnight before addition of 20 µl equilibrated Dynabeads Protein G. After 4 hours of rotation at 4˚C, the supernatant was removed and the beads were washed twice with 700 µl FA lysis buffer, once with 700 µl ChIP wash buffer and once with 700 µl TE. DNA was eluted from the beads by addition of 100 µl ChIP elution buffer and incubation at 65˚C for 10 min. DNA was then processed as described [70] and used for qPCR as described above.

## REB1-ChIP

*REB1-TAP* strains were grown to an $OD_{600}$ of 0.6–0.8. ChIP was performed as described in [70] using Pan Mouse IgG Dynabeads (Invitrogen).

## TERRA-RT-qPCR

Overnight cultures were diluted to $OD_{600}$ of 0.2–0.3, grown for 3–5 hours and 10–20 $OD_{600}$ units of cells were harvested in mid-logarithmic growth phase. Total RNA was isolated with a modified RNA extraction protocol [71].

Cell pellets were resuspended in 400 µl AE buffer (50 mM Na-acetate pH 5.3, 10 mM EDTA), with 10% SDS added to a final concentration of 0.9%. After hot acid phenol extraction, the supernatant was transferred to a new microfuge tube and 500 µl 25:24:1 phenol:chloroform:isoamyl alcohol equilibrated with RNA buffer (0.5 M NaCl, 200 mM Tris-HCl, 10 mM EDTA) were added. The total nucleic acid extraction was performed as described until pelleted.

The pellet was resuspended in 100 µl DNase mixture containing 6 units of DNase (NEB) and incubated 30 min at 37˚C. 50 µg RNA was purified with the RNeasy MinElute Cleanup Kit (Qiagen) according to manufacturer instructions, and eluted with 40 µl ddH$_2$O. DNase mixture was added to a total volume of 100 µl with 6 units of DNase (NEB) and incubated 30 min at 37˚C. After a second purification with the RNeasy MinElute Cleanup Kit (Qiagen), DNase mixture was added to a total volume of 100 µl with 6 units of TURBO DNase (Invitrogen) and incubated 30 min at 37˚C to remove any residual trace amounts of DNA. The RNA was purified a third time with the RNeasy MinElute Cleanup Kit (Qiagen) and finally eluted in 30 µl ddH$_2$O. RNA concentration was calculated as described previously.

Reverse transcription was performed on 3.0 µg of triple DNase-treated RNA. First, an annealing premixture containing the RNA, dNTPs and the TERRA-RT (oBL207) and ACT1-RT primers (see S3 Table) were heated to 90˚C for 1 min, then cooled with a rate of 0.8˚C/s down to 55˚C. Next, the +RT and–RT reaction mixtures were added to the tubes. The +RT mixture contains SuperScript III Reverse Transcriptase and First Strand buffer (Invitrogen). The total reaction was 20 µl and contains a final concentration of 0.5 mM dNTPs, 0.5 µM TERRA-RT primer and 0.2 µM ACT-RT primer. The mix was incubated at 55˚C for 60 min followed by 15 min at 70˚C.

Next, the +RT and–RT reactions were diluted 2.5x to a total volume of 50 µl and used as template for quantitative PCR. Samples showing less than 3.0 ΔCt between +RT and–RT were excluded from further analysis.

## 5′RACE

Total RNA was isolated as mentioned above and used for full-length, RNA Ligase-Mediated Rapid Amplification of 5′ cDNA Ends (RLM-RACE) with the GeneRacer Kit (Invitrogen, Catalog no. L1502-01). Manufacturer instructions were followed with minor modifications as follows:

Reverse transcription was performed with oBL207 to ensure enrichment of X-element containing TERRAs.

Complementary DNA ends were amplified in the first round of PCR with the provided GeneRacer 5′ Nested Primer and reverse primer STR-F (see S3 Table) using standard *Taq* polymerase. The resulting products were separated on a 1.5% agarose gel, extracted from the gel and subsequently used for ligation into the pGEM-T Vector system I (Promega, Madison, Wisconsin, USA) as per manufacturer instructions. Inserts were sequenced in both directions.

For the TERRA transcriptional start site in TEL6R, the first round of PCR was performed using GeneRacer 5′ Primer and oBL207 using standard *Taq* polymerase. Two μl of 20-fold diluted PCR products were used for a second round of PCR using GeneRacer 5′ Nested Primer and oBL259 (see S3 Table). The resulting products were processed as described above.

## Transcription termination assay

A *cup1 reb1*<sup>ts</sup> strain transformed with p*CUP1* or p*TEL3L-X-CUP1* (containing *TEL3L-X* in both orientations) was grown overnight in SC–histidine + 2% glucose, diluted to $OD_{600} = 1.0$. A 10-fold serial dilution was spotted on SC–histidine + 2% galactose plates with different concentrations of $CuSO_4$ and incubated 3 days at 30˚C before images were acquired.

## Statistical analysis

Statistical analysis used the two-tailed unpaired *t*-test (*$p < 0.05$, **$p < 0.01$, ***$p < 0.001$ and ****$p < 0.0001$).

## Supporting information

**S1 Table. DANPOS list of nucleosomes.**
(XLSX)

**S2 Table. Yeast strains used in this study.**
(XLSX)

**S3 Table. Oligonucleotides used in this study.**
(XLSX)

**S4 Table. Plasmids used in this study.**
(XLSX)

**S1 Fig. Chromatin features of *WT* and *sir2* strains at the telomeres.** Illustrations of a 3000 bp region of the indicated telomeres in the Integrative Genome Viewer (IGV). Shown are tracks generated from MNase-seq tracks of *WT* (grey), *sir2* (orange) and the DANPOS-calculated difference between *sir2* and *WT* (black), including FDR values when applicable. Reb1 consensus sites are denoted in blue and Reb1 occupancy levels (ChIP-exo) are shown in dark red. Genes are shown in the bottom track in grey.
(PDF)

**S2 Fig. Chromatin features of *WT* and *sir2* strains at the telomeres, continued from S1 Fig.**
(PDF)

**S3 Fig. Chromatin features of *WT* and *sir2* strains at the telomeres, continued from S2 Fig.**
(PDF)

**S4 Fig. Chromatin features of *WT* and *sir2* strains at the telomeres, continued from S3 Fig.**
(PDF)

**S5 Fig. Chromatin features of *WT* and *sir2* strains at the telomeres, continued from S4 Fig.**
(PDF)

**S6 Fig. Chromatin features of *WT* and *sir2* strains at the telomeres, continued from S5 Fig.**
(PDF)

**S7 Fig. Chromatin features of *WT* and *sir2* strains at the telomeres, as in S1–S6 Fig.**
MNase-seq data was aligned to W303 genome.
(PDF)

**S8 Fig. Chromatin features of *WT* and *sir2* strains at the telomeres, continued from S7 Fig.**
(PDF)

**S9 Fig. The nucleosome occupancy change is independent of cell type, but depends on the Sir-complex.** (**A**) MNase-qPCR of strains deleted for the promoter regions of all 3 mating type loci (*mat*a$\Delta p$ *hml*a$\Delta p$ *hmr*a$\Delta p$) with or without *SIR2*. (**B**) MNase-qPCR of *WT*, *sir2* and *sir3* strains. (**C**) MNase-qPCR of strains expressing either *WT* Histone 4 or mutant alleles mimicking acetylated/deacetylated lysine residue 16. All data are shown as mean ± SEM (n = 3, *p$<$0.05, **p$<$0.01, ***p$<$0.001, ****p$<$0.0001, ns = not significant).
(PDF)

**S10 Fig. The *reb1$^{ts}$* allele encodes a stable protein, but cannot support viability at 37˚C.** (**A**) Ten-fold serial dilutions of the indicated strains were spotted on synthetic complete media and incubated at the indicated temperatures for 3 days. (**B**) Western blot analysis of protein extracts from strains containing the indicated Myc-tagged *REB1* alleles at different temperatures and time points using an anti-Myc antiserum (9E11). Lower panel shows Ponceau S staining of the membrane.
(PDF)

**S11 Fig. Overexpression of Reb1 destabilizes the STR-D nucleosome and determination of TERRA TSS by 5′RACE.** (**A**) MNase-qPCR of *WT* and *sir2* strains carrying both endogenous *REB1* as well as a plasmid-borne copy under the control of the *GAL1* promoter. Strains were grown in media containing either 2% glucose (no overexpression) or 2% galactose (overexpression of Reb1, dotted columns). Glu: Glucose, Gal: Galactose (**B**) Nested PCR of 5′RACE performed either with (+) or without (-) the addition of Tobacco Acid Pyrophosphatase (TAP) to remove the 5′ cap structure of the RNA in the indicated strains to map the TSS of 6R TERRA.
(PDF)

**S12 Fig. Preliminary model for the role of Reb1 and Sir-proteins in regulating X-element nucleosome occupancy and TERRA transcription.** (**A**) In the *WT* strain, Reb1 binds to several binding sites in the X-elements, but binds less to the STR-D loci due to competition with the Sir-complex that stabilizes the STR-D nucleosome. This leads to low TERRA steady-state levels as both transcriptional initiation and elongation are repressed. (**B**) In the *sir2* strain, the STR-D nucleosome is lost and Reb1 binding increases. TERRA levels rise due to the increased accessibility for the transcriptional machinery, but Reb1 binding still restricts transcriptional elongation of TERRAs. (**C**) In the *reb1$^{ts}$ sir2* double mutant, the nucleosome is reinstated and TERRA transcription increases synergistically (bottom). See text for further details.
(PDF)

## Acknowledgments

We thank J Warner, J Rine, P San-Segundo, A Byström for yeast strains, and D Libri for yeast strains and plasmids. We thank B Luke and S Misino for technical advice on TERRA detection.

We thank C Andréasson for technical advice on generation of the *reb1^ts* allele. We thank A Musli for generation of epitope-tagged Reb1-alleles and analysis of Reb1 stability.

## Author Contributions

**Conceptualization:** Stefan U. Åström.

**Data curation:** Stefanie L. Bauer, Thomas N. T. Grochalski, Agata Smialowska.

**Formal analysis:** Stefanie L. Bauer, Thomas N. T. Grochalski, Agata Smialowska, Stefan U. Åström.

**Funding acquisition:** Stefanie L. Bauer, Stefan U. Åström.

**Investigation:** Stefanie L. Bauer, Thomas N. T. Grochalski.

**Methodology:** Thomas N. T. Grochalski, Agata Smialowska.

**Project administration:** Stefan U. Åström.

**Resources:** Stefan U. Åström.

**Software:** Stefanie L. Bauer, Agata Smialowska.

**Supervision:** Stefan U. Åström.

**Validation:** Stefanie L. Bauer, Thomas N. T. Grochalski, Stefan U. Åström.

**Visualization:** Stefanie L. Bauer, Thomas N. T. Grochalski, Agata Smialowska.

**Writing – original draft:** Stefanie L. Bauer, Stefan U. Åström.

**Writing – review & editing:** Stefanie L. Bauer, Thomas N. T. Grochalski, Agata Smialowska, Stefan U. Åström.

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
