## [Decision Letter · Decision Letter 0]

27 Jul 2022

Dear Dr Åström,

Thank you very much for submitting your Research Article entitled 'Sir2 and Reb1 antagonistically regulate nucleosome occupancy in subtelomeric X-elements and repress TERRAs by distinct mechanisms' to PLOS Genetics.

The manuscript was fully evaluated at the editorial level and by independent peer reviewers. The reviewers appreciated the attention to an important topic but identified some concerns that we ask you address in a revised manuscript. We therefore ask you to modify the manuscript according to the review recommendations. Your revisions should address the specific points made by each reviewer.

[LINK]

Yours sincerely,

Jin-Qiu Zhou

Academic Editor

PLOS Genetics

Gregory P. Copenhaver

Editor-in-Chief

PLOS Genetics

Reviewer's Responses to Questions

**Comments to the Authors:**

Reviewer #1: In this study, Bauer et all use the S. cerevisiae to study the chromatin organization at the telomere in cells deficient for the founding member of the sirtuin KDAC family Sir2, and/or the Reb1 transcription factor. The authors use a combination of genetic analysis and MNase-seq and MNase-ChIP-qPCR to demonstrate that sir2 mutant cells have a change in nucleosome positioning in the X elements of telomeres. In particular, sir2 mutation seems to destabilize a nucleosome in the STR-D region of the X elements, and this is dependent on position of the X-element at the telomere. Absence of the Reb1 transcription factor rescues the nucleosome positioning. Interestingly, absence of Reb1 and Sir2 both result in increased TERRA transcription, whose TSS seems to overlap with the relevant nucleosome. Altogether the work points to a model where Sir2 and Reb1 compete to stabilize/destabilize the STR-D nucleosome while impacting TERRA transcription via a distinct mechanism.

I liked this paper. The experiments are well conceived and well controlled, and the data looks clean. I also think that the Reb1 temperature sensitive allele will be a valuable tool for other groups to further work in the field. The writing is excellent ,and overall the manuscript was a pleasure to read.

Comments:

A) The authors state “In the W303 strains used by us, TEL1R, 7L, 13R and 14R have deletions in the subtelomeric regions compared to the reference strain (S288C), partly explaining the lack of read”.

This got me a bit worried. How can the authors be sure that other conclusions are not impacted by mapping onto S288C genome?

There is a full reference w303 genome:  Whole-Genome Sequence and Variant Analysis of W303, a Widely-Used Strain of Saccharomyces cerevisiae. Matheson K, Parsons L, Gammie A.

G3 (Bethesda). 2017 Jul 5;7(7):2219-2226. doi: 10.1534/g3.117.040022.

Which makes it a bit curious that the authors did their alignments to the S288C genome, might be a good idea for them to double check that their conclusions hold with the W303 reference genome.

B) The model proposed by the authors suggests a competition between Reb1 and Sir2. The authors show increased Reb1 binding in Sir2 mutants, but they should also test the opposite: ie, is there increased Sir2 binding in the reb1-ts strain. Alternatively, the authors should clearly define this as a remaining caveat/future direction and modify the relevant text.

C) Line 266: “We next set out to test the effect of depleting Reb1 on nucleosome occupancy. To this end, a temperature-sensitive allele of Reb1 was generated (see materials and methods).”

Suggest change to:

“We next set out to test the effect of depleting Reb1 on nucleosome occupancy. Since Reb1 is essential, a temperature-sensitive allele of Reb1 was generated (see materials and methods).”

D) The authors could include an overall model for the Reb1 and Sir2 function in nucleosome positioning and TERRA transcription.

E) I recommend switching the order of Figure 5A and Figure 5B.

F) In the discussion the authors mention that Sir2 could be targeting other lysine residues, but could clearly mention the possibility that Sir2 is working via targeting of both histone and non-histone proteins.

Reviewer #2: Telomeres and subtelomeres in Saccharomyces cerevisiae are bound by a number of proteins including Rap1 and its cofactors Rif1-Rif2 and the Sir complex at telomeres/subtelomeres, and Abf1, Tbf1 and Reb1 at subtelomeres. They not only protect the extremities from being recognized as double-strand breaks, but also contribute to the regulation of many aspects of telomere biology, including telomerase recruitment, chromatin structure/status and TERRA expression. In this manuscript, Bauer et al. investigate and dissect the functions of the histone deacetylase Sir2 and the transcription factor Reb1 in nucleosome positioning and occupancy in the subtelomeric X-element. They use MNase-seq and a diverse set of complementary approaches to show that for a majority of subtelomeres, Sir2 is important to maintain a nucleosome precisely positioned at the STR-D element and Reb1 tends to have the opposite effect. However, both factors repress TERRA expression, but through different mechanisms.

Overall, as far as I can tell, this work is well executed, the results are very convincing and the conclusions strongly supported by the experiments. I have a list of comments/questions and minor issues that could be optionally addressed to improve the manuscript.

1. I appreciate that the authors addressed the issue of the derepression of the mating-type loci HML/HMR in the sir2∆ mutant in Fig. S7A.

2. Given that Y’ elements share a high percentage of sequence identity, how were the reads accurately mapped to specific Y’ elements?

3. Is there an accurate W303 genome that the authors could have used to map the reads, instead of R64-1-1? They could have had a better view of subtelomeres 1R, 7L, 13R and 14R.

4. The sir2∆ results in Fig. 2B do not seem to be referenced in the text.

5. Lines 228-230 and 463-465: based on the result using the mutant sir2-N345A, the authors suggest that the catalytic activity of Sir2 might be important for nucleosome stabilization. However, Armstrong et al. (ref 34) showed that this mutant fails to properly localize to the telomere. The reduction of nucleosome occupancy in this mutant might not be directly due to a defect in the catalytic activity, but rather due to a decreased recruitment of Sir2.

6. Lines 144-146: although it is fair that the authors wish to present their results for nucleosome regulation at other loci in another publication, it would be nice to know if the mechanism shown here is specific to subtelomeres or more general. I do not know whether referring to unpublished data (which is no longer a common practice) or showing a figure without revealing too much would be a solution.

7. Fig. S8B: can the authors comment on the slight change in mobility between Reb1 and Reb1ts?

8. Fig. 4C: the transcription termination assay demonstrates that the X-element can terminate transcription. However, whether this property is Reb1-dependent was not really assessed in a convincing way. The authors used the reb1ts mutant at permissive temperature, arguing that it has a partial loss of function, and showed that it is less efficient than wild-type to terminate transcription. Ideally, the authors could perform the assay with reb1ts at a semi-restrictive temperature (or over a range of temperatures), so that cells would still grow rather well without Cu2+ but would show a stronger growth defect in the presence of Cu2+. Alternatively, the assay could be performed with a X-element mutated for its Reb1 binding site(s).

9. Lines 289-290: Dr Krallis (from the Wellinger lab) did experiments testing Reb1’s role in regulating TERRA. They are reported in her PhD thesis: https://savoirs.usherbrooke.ca/bitstream/handle/11143/17175/Krallis_Alexandra_MSc_2020.pdf?sequence=1&isAllowed=y

But their work has not been published in a peer-reviewed journal yet, to my knowledge.

10. It is not straightforward to conclude about the lack of correlation between telomere length and TERRA expression using mutants as examples, due to potential indirect effects of the mutants.

11. In most graphs, statistical tests are shown for comparison against the wild-type condition, but sometimes additional tests can be useful, for example in Fig. 5A “amplicon 2” between URA3::TEL3L-X and URA3::TEL3L-X sir2∆. By the way, can the authors explain this observation of higher nucleosome occupancy in the absence of Sir2 at the URA3 locus?

12. A prediction regarding nucleosome occupancy affecting TERRA expression is that at an X-element where sir2∆ does not show nucleosome depletion, e.g. 5R, TERRA should not be increased upon SIR2 deletion or Reb1 delocalization (using reb1ts at restrictive temperature). Did the authors test this prediction?

13. Line 25: “Reb” should be “Reb1”.

14. Lines 95-96: for the discovery of TERRA, Azzalin et al. 2007 Science should be cited in my opinion, even though the work was done in mammalian cells.

15. Fig. 1C: maybe the data points corresponding to the same chromosome extremity in WT vs sir2∆ could visually linked with a straight line for example? The points corresponding to X-core of to STR-D could also be distinguished using different colors.

Reviewer #3: In the manuscript by Bauer et al, the authors demonstrate that subtelomeric nucleosomes -specifically at the STR-D elements, are stabilized by Sir2/Sir3. They suggest that this stabilization occurs by blocking Reb1 from binding to its consensus sequences. Indeed, in a sir2 deletion strain nucleosomes are depleted as demonstrated by MNase-seq and verified by ChIP qPCR, the further inactivation of Reb1, through a reb1-ts allele, restores nucleosome occupancy, whereas the overexpression of Reb1 exacerbates the nucleosome depletion phenotype. The authors demonstrate that this has an effect on TERRA levels as reb1-ts strains have increased TERRA levels and demonstrate synergistic increases with loss of Sir2. The authors use a plasmid-based reporter assay to conclude that Reb1 is likely acting as a transcriptional terminator (roadblock). The manuscript is well done and shows some interesting insights into the regulation of telomeric transcription and chromatin structure….unfortunately it lacs functional readouts, which should be possible as some TERRA functions are known. Since TERRA and TERRA R-loops have been shown to be important during replicative senescence, I would like to suggest one or two experiments that may be interesting to try and may strengthen the manuscript even further. I also have a few minor criticisms that I would like to see addressed.

1. TERRA R-loops have been shown to delay the onset of replicative senescence. Can the authors test whether the reb1-ts strain delays replicative senescence, as there should be more TERRA R-loops present due to the lac of termination. Testing Reb1 overexpression as well would be interesting in this context.

2. TERRA is regulated in a cell cycle dependent manner in both yeast and human cells as demonstrated by the Lingner and Luke labs. Does this correspond at all with Reb1 occupancy? i.e. is Reb1 at telomeres in G1, but absent in early S phase when TERRA gets transcribed (have a look at Graf et al, alpha factor and HU experiments)

3. I am a bit worried about the temperature effects on TERRA levels as temperature drastically affects TERRA levels. Can you also see the nucleosome effects/TERRA effects with a Reb1 degron, hence avoiding temp shifts.

4. Finally, it the increase in telomere length due to telomerase or HR?

**Have all data underlying the figures and results presented in the manuscript been provided?**

Reviewer #1: Yes

Reviewer #2: **No: **I was not able to access the MNase-seq data on GEO (GSE195972). But it seems the data is already deposited at least.

Reviewer #3: Yes

PLOS authors have the option to publish the peer review history of their article (what does this mean?). If published, this will include your full peer review and any attached files.

Reviewer #1: No

Reviewer #2: No

Reviewer #3: No

---

## [Decision Letter · Decision Letter 1]

8 Sep 2022

Dear Dr Åström,

We are pleased to inform you that your manuscript entitled "Sir2 and Reb1 antagonistically regulate nucleosome occupancy in subtelomeric X-elements and repress TERRAs by distinct mechanisms" has been editorially accepted for publication in PLOS Genetics. Congratulations!

Please note that Reviewer #1 has one small optional suggestion (see below) that you may want to consider as you prepare you final draft for the production team (the editorial teal will not need to re-evaluate).

Yours sincerely,

Jin-Qiu Zhou

Academic Editor

PLOS Genetics

Gregory P. Copenhaver

Editor-in-Chief

PLOS Genetics

Comments from the reviewers (if applicable):

Reviewer's Responses to Questions

**Comments to the Authors:**

Reviewer #1: My concerns, which centred mostly on the W303 versus S288C alignments, have been addressed. Again, congrats on a nice story.

Something for the authors to consider as a small edit:

In the discussion authors state:

"The work presented herein agrees well with this observation and shows that this triple nucleosome structure depends on Sir2". Perhaps "at most telomeres" should be added tot he end of this sentence. The authors could also speculate why some telomeres e.g. 5L, 8L and 16R do not show the same changes as other telomeres. I leave it to the authors to decide whether these changes might improve the discussion and I recommend accepting the manuscript either way and do not need to evaluate the manuscript again.

Reviewer #2: This reviewer had no major concern to begin with and the authors have addressed most of the remaining comments. The manuscript looks very solid and convincing.

**Have all data underlying the figures and results presented in the manuscript been provided?**

Reviewer #1: Yes

Reviewer #2: Yes

PLOS authors have the option to publish the peer review history of their article (what does this mean?). If published, this will include your full peer review and any attached files.

Reviewer #1: No

Reviewer #2: No

**Data Deposition**

http://datadryad.org/submit?journalID=pgenetics&manu=PGENETICS-D-22-00717R1

**Press Queries**

---

## [Editor Report · Acceptance letter]

19 Sep 2022

PGENETICS-D-22-00717R1 

Sir2 and Reb1 antagonistically regulate nucleosome occupancy in subtelomeric X-elements and repress TERRAs by distinct mechanisms 

Dear Dr Åström, 

We are pleased to inform you that your manuscript entitled "Sir2 and Reb1 antagonistically regulate nucleosome occupancy in subtelomeric X-elements and repress TERRAs by distinct mechanisms" has been formally accepted for publication in PLOS Genetics! Your manuscript is now with our production department and you will be notified of the publication date in due course.

With kind regards,

Zsofia Freund

PLOS Genetics

On behalf of:
